# A Review of Syndromic Forms of Obesity: Genetic Etiology, Clinical Features, and Molecular Diagnosis

**DOI:** 10.3390/cimb47090718

**Published:** 2025-09-03

**Authors:** Anam Farzand, Mohd Adzim Khalil Rohin, Sana Javaid Awan, Zubair Sharif, Adnan Yaseen, Abdul Momin Rizwan Ahmad

**Affiliations:** 1School of Nutrition and Dietetics, Faculty of Health Sciences, Universiti Sultan Zainal Abidin (UNISZA), Gong Badak Campus, Kuala Nerus 21300, Malaysia; si4217@putra.unisza.edu.my (A.F.); mohdadzim@unisza.edu.my (M.A.K.R.); 2Institute of Molecular Biology and Biotechnology (IMBB), The University of Lahore, Lahore 54500, Pakistan; sana.javaidawan@yahoo.com; 3Department of Medical Laboratory Technology, Faculty of Allied Health Sciences, Superior University, Lahore 54000, Pakistan; zubair.sharif@superior.edu.pk; 4Shaikh Zayed Medical Complex, Lahore 54000, Pakistan; adnan.yaseen97@gmail.com; 5Department of Health Sciences, University of York, York YO10 5NH, UK; 6Department of Human Nutrition and Dietetics, NUST School of Health Sciences, National University of Sciences & Technology (NUST), Sector H-12, Islamabad 44000, Pakistan

**Keywords:** syndromic obesity, genetic etiology, molecular diagnosis, obesity-related syndromes, clinical features of obesity, rare genetic disorders, monogenic obesity

## Abstract

Background: Syndromic forms of obesity are uncommon, complicated illnesses that include early-onset obesity along with other clinical characteristics such as organ-specific abnormalities, dysmorphic symptoms, and intellectual incapacity. These syndromes frequently have a strong genetic foundation, involving copy number variations, monogenic mutations, and chromosomal abnormalities. Methods: Using terms like “syndromic obesity,” “genetic diagnosis,” and “monogenic obesity,” a comprehensive literature search was conducted to find articles published between 2000 and 2025 in PubMed, Scopus, and Web of Science. Peer-reviewed research addressing the clinical, molecular, or genetic aspects of syndromic obesity were among the inclusion criteria. Conference abstracts, non-English publications, and research without genetic validation were among the exclusion criteria. The whole genetic, clinical, diagnostic, and therapeutic domains were thematically synthesized to create a thorough, fact-based story. Research using chromosomal microarray analysis (CMA), whole-exome sequencing (WES), next-generation sequencing (NGS), and new long-read sequencing platforms was highlighted. Results: Despite the fact that molecular diagnostics, especially NGS and CMA, have made tremendous progress in identifying pathogenic variants, between 30 and 40 percent of instances of syndromic obesity are still genetically unexplained. One significant issue is the variation in phenotype across people with the same mutation, which suggests the impact of environmental modifiers and epigenetic variables. In addition, differences in access to genetic testing, particularly in areas with limited resources, can make it difficult to diagnose patients in a timely manner. Additionally, recent research emphasizes the possible contribution of gene–environment interactions, gut microbiota, and multi-omic integration to modifying disease expression. Conclusions: Syndromic obesity is still poorly understood in a variety of groups despite significant advancements in technology. Multi-layered genomic investigations, functional genomic integration, and standardized diagnostic frameworks are necessary to close existing gaps. The development of tailored treatment plans, such as gene editing and focused pharmaceutical therapies as well as fair access to cutting-edge diagnostics are essential to improving outcomes for people with syndromic obesity.

## 1. Introduction

Monogenic obesity syndromes offer key insights into energy homeostasis and appetite regulation [1]. Up to 25% of patients with early-onset morbid obesity carry mutations in at least one described gene [2]. Genetic influence also extends to common polygenic obesity, challenging the assumption that it is purely multifactorial. This review synthesizes the current understanding of syndromic obesity, with emphasis on genetic etiology and diagnostic strategies [3]. Obesity is a multifactorial disorder with cardiometabolic, endocrine, and neurocognitive effects [4]. Genetic factors account for up to 70% of BMI variability [5]. Several mutations involved in syndromic obesity have been used for therapeutic intervention, for example, by using MC3R agonist [Setmelanotide], while no affordable or effective therapy exists for others [6]. Here, we outline major syndromic forms of obesity, their genetic causes, and hallmark clinical features [7]. Histological or imaging studies of adipose tissue can further clarify gene-specific effects and aid molecular diagnosis [8].

## 2. Literature Search Strategy and Scope of Review

We searched PubMed, Scopus, and Web of Science (2000–2025) using “syndromic obesity,” “genetic diagnosis,” and “monogenic obesity.” Eligible studies were peer-reviewed and addressed clinical, molecular, or genetic aspects; non-English articles, conference abstracts, and studies lacking genetic validation were excluded. Data were synthesized across genetic, clinical, diagnostic, and therapeutic domains.

## 3. Genetic Etiology, Clinical Features, and Diagnostic Approaches

### 3.1. Overview and Classification of Syndromic Obesity

The complexity of human obesity is emphasized by the extensive number of genetic loci contributing to common forms. Figure 1 compares the monogenic and polygenic obesity regarding genetic influence, mutation type, and environmental contribution [9].

Monogenic obesity arises from rare, high-penetrance single-gene mutations with minimal environmental influence, whereas polygenic forms result from many low-effect variants strongly modified by the environment.

The rapid progression in genome sequencing has expanded the recognized syndromes to over [161 syndromes as of the end of 2014] [10]. Syndromic obesity is characterized by early-onset obesity co-occurring with neurodevelopmental, dysmorphic, and endocrine abnormalities [11]. Phenotypic characterization aids molecular diagnosis, especially in patients with intellectual disability, behavioral disorders, specific craniofacial features, and endocrine and metabolic abnormalities [12,13].

Gaining insight into syndromic obesity improves our understanding of the important hormonal and neuroendocrine systems that control energy balance. These understandings are essential for both guiding customized treatments and early diagnosis. Current treatment approaches for congenital and genetically induced obesity include surgical, pharmaceutical, and lifestyle treatments, emphasizing the multifaceted character of care [14,15]. Figure 2 illustrates these diverse therapeutic approaches.

### 3.2. Genetic Etiology (Monogenic and Chromosomal Syndromes)

Although obesity is highly heritable [9], environmental factors such as sedentary lifestyles and hypercaloric diets contribute significantly to its rising prevalence [16]. Some obese individuals present with genetic syndromes or chromosomal abnormalities [primarily submicroscopic deletions/duplications] due to structural variations [17]. While over 700 loci are linked to common obesity, only one-third involve hypothalamic genes, suggesting most act peripherally [12,18]. Syndromic forms are typically monogenic, often autosomal dominant, and paternally inherited, with many genes imprinted, maternally expressed, and epigenetically regulated [1,19,20]. Around one-third involve submicroscopic deletions and another third single-gene mutations, most affecting hypothalamic functions [21,22,23,24].

#### Monogenic Syndromes Associated with Obesity

FGFR1 defects are linked to hypogonadotropic hypogonadism, hyperleptinemia, dental anomalies, and altered AKT1–mTOR signaling [25,26]. Hormonal replacement therapy may improve clinical outcomes in affected patients [27].

An overview of the key milestones and population-specific discoveries in obesity genetics is illustrated in Figure 3. The lower panel shows essential events from 1994 to 2020, starting with the cloning of the leptin [LEP] gene and moving on to the discovery of critical candidate genes [such as LEPR, MC4R, POMC, BDNF, SH2B1, and ADCY3] through both candidate gene and genome-wide linkage studies. The upper panel shows the number of obesity-related loci found by genome-wide association studies [GWAS] from 2007 to 2020. The contributions are broken down by ancestry [European, Asian, African, and Other]. The graph shows a massive rise in discovered loci, mostly from European populations. This indicates that we are learning more about the genetic architecture of obesity and that genetic studies need to include more diverse ancestral groups.

SAX1 deficiency is a rare genetic syndrome characterized by exocrine system defects, sensorineural hearing loss, ingestive difficulties, vertigo, and obesity. Reported cases also demonstrate persistent hyperphagia and obesity, underscoring the importance of comprehensive genetic and clinical evaluation [28,29]. This case highlights the phenotypic heterogeneity of syndromic obesity and the importance of comprehensive genetic evaluation. While syndromic types of obesity are less common than polygenic obesity, they nonetheless include a diverse range of conditions that can appear in children and adults. Various factors can cause them, most frequently chromosomal or monogenic defects, but secondary endocrine dysfunction is another possibility. Neurodevelopmental, dysmorphic, or metabolic characteristics are often present in monogenic and chromosomal disorders, which are characterized by high-impact genetic variants that affect the hypothalamic regulation of hunger and energy balance. On the other hand, primary hypothyroidism, Cushing syndrome, hypothalamic abnormalities (for example, from cranial pathology), or hyperinsulinism as a result of Beckwith–Wiedemann syndrome can all lead to endocrine-mediated obesity. Understanding these etiological differences is crucial to direct targeted molecular diagnostics, inform prognosis, and choose the best management approaches [30,31].

Even though the prevalence of obesity is rising globally—often referred to as an “epidemic”—many patients’ therapy results are still below ideal. Particularly, intragastric balloon implantation and oral anti-obesity medications have been shown to have poor long-term efficacy [32]. These restrictions underscore the urgent need for precision-based approaches catered to the underlying genetic and phenotypic profiles of impacted individuals by highlighting a significant disconnect between the scope of the public health issue and the efficacy of currently available therapies.

Genetic obesity accounts for <5% of general clinic cases but exceeds 10% of severe early-onset cases [1,9], often with strong family history, near-complete penetrance, and more than 20 known causal genes across approximately 30 single-gene syndromes [18,33,34]. Table 1 summarizes the major syndromic forms, which may be hypothalamic in origin or associated with hyperphagia, cognitive impairment, and neuroanatomical changes [14,35,36]. Persistent hyperphagia is a common and often under-recognized as pathological.

### 3.3. Clinical Features and Phenotypic Variability

Each disorder presents a distinct array of clinical characteristics, the intensity of which may vary depending on the specific gene involved and its mode of inheritance [dominant, recessive, or X-linked] [37]. Recognition may be challenging due to phenotypic complexity [38,39].

Visceral adiposity is a hallmark in both pediatric and adult syndromic obesity cases [40]. In most syndromes, obesity develops rapidly before age 6, often with impaired satiety and hyperphagia [14].

Other features may also include hypotonia, behavioral and intellectual disability, and multiple somatic malformations [e.g., dysmorphisms, extremities abnormalities, genital abnormalities, strabismus, heart defects, alopecia, and skin and hematological disturbances, including neoplasia] [41]. The heterogeneity of clinical manifestations is also evident in LEP and LEPR deficiencies, typically presenting with severe hyperphagia, rapid evolution of obesity, and endocrine abnormalities [42]. A thorough endocrine, genetic, and metabolic assessment is essential for molecular diagnosis [10]. These multifaceted contributors to syndromic obesity are summarized in Figure 4.

#### Common Clinical Manifestations

Clinical features include obesity, hyperphagia, and cognitive impairment (from mild attention deficits to severe intellectual disabilities as well as the inability to acquire language) [11,43]. The onset may be present from early life to adulthood [44]. Endocrinological anomalies may include hypothyroidism, hypogonadotropic hypogonadism/associated cryptorchidism, osteoporosis, low IGF-1 levels, growth hormone deficiency, hyperinsulinemia, and Type 2 diabetes mellitus [45].

Rare cases, often due to deletions/duplications and small point mutations, may exhibit a series of morphological abnormalities. These include more complex, intense-period, multiplied craniofacial and limb malformations [midface hypoplasia, ocular hypertelorism, micro- or retrognathia, short neck, brachydactyly, and mild syndactyly] [46,47]. Syndromic presentations vary in severity, ranging from subtle dysmorphism to pronounced multisystem involvement [48]. In very few cases, a mutation in a single gene has been reported primarily with the obesity phenotype [21].

The figure above gives a complete description of the multiple causes of syndromic obesity. It divides contributing factors into four categories: cognitive deficiencies, such as attention span issues and language acquisition difficulties; genetic mutations, such as deletions, duplications, and point mutations; endocrinological anomalies, including hyperinsulinemia and hypothyroidism; and morphological abnormalities, such as limb and craniofacial malformations. Together, these interrelated elements highlight the intricacy of syndromic obesity and the several molecular mechanisms that influence its development [49].

This table shows a synopsis of the obesity types that have been studied the most extensively. There is a description of the syndrome, its mechanism of inheritance, the genes or chromosomes that are impacted, and the characteristic clinical characteristics considered to be the condition’s hallmarks. Signs and symptoms that patients may experience include gaining weight quickly, eating excessively, experiencing delays in development, abnormal facial features, and abnormalities in specific organs.

### 3.4. Molecular Diagnosis Approaches (Including NGS/WES/WGS)

Several syndromic forms of obesity, such as Prader–Willi and Bardet–Biedl syndromes, are characterized by early-onset central adiposity, hypotonia, intellectual disability, dysmorphic features, and endocrine abnormalities [50]. These hallmarks guide target genetic testing [51,52]. Molecular diagnosis increasingly relies on next-generation sequencing (NGS), which enables high-throughput screening of both established and novel obesity-related genes, thereby improving diagnostic accuracy while still presenting certain limitations [53,54]. Whole-exome sequencing [WES] clarifies the genetic cause in around one-third of patients, often avoiding sequential single-gene testing and revealing non-genetic factors. Recent advances in molecular diagnostics, particularly whole-exome sequencing (WES), whole-genome sequencing (WGS), and next-generation sequencing (NGS), have clarified the genetic cause in up to one-third of patients, achieving diagnostic yields of ~60% in suspected cases and deepening our understanding of genetic and phenotypic diversity in syndromic obesity [55,56]. They are rapidly becoming the mainstay of diagnosis, enabling individualized treatment planning and discovery of novel disease genes [57,58].

### 3.5. Landmark Studies and Diagnostic Yield

Key studies have significantly influenced our current understanding of syndromic obesity:Leptin-melanocortin pathway: Farooqi and O’Rahilly identified mutations in LEPR, MC4R, and POMC, establishing hypothalamic regulation in monogenic obesity [59].Targeted Therapy: FDA approval of Setmelanotide [MC4R agonist] for specific genetic forms of obesity underscore the therapeutic relevance of gene diagnosis [60].WES cohorts: Large studies found actionable mutations in 25–30% of early-onset severe obesity cases, often missed with traditional Sanger sequencing [61].NGS panels: Mutation detection rates up to 65% in clinically suspected cases emphasizing the power of modern sequencing [62].

These advances link molecular discovery to improved diagnosis, precision therapies, and functional genomics.

New developments in the pharmaceutical management of obesity have brought in promising drugs that influence central appetite regulation and gut-derived incretin pathways. GLP-1 receptor agonists, including semaglutide and liraglutide, have shown notable effectiveness in reducing body weight by increasing satiety, delaying stomach emptying, and boosting insulin sensitivity. Building on this basis, phase III clinical trials have demonstrated the superior results of dual agonists such as tirzepatide, which target both GLP-1 and GIP receptors, showing synergistic effects on obesity and glycemic management. Furthermore, triple agonists, such as retatrutide (GLP-1/GIP/glucagon receptor agonist), offer improved energy expenditure, appetite reduction, and lipid profile improvement by simultaneously engaging several receptors [59]. This represents a new frontier in metabolic control. Crucially, the treatment landscape for syndromic and monogenic types of obesity is changing from general metabolic control to precision therapies that target specific pathways. For individuals who are obese because of POMC, LEPR, or PCSK1 deficits, the FDA-approved MC4R agonist Setmelanotide is now accessible. It has shown remarkable effectiveness in lowering hyperphagia and achieving long-term weight loss without causing serious cardiovascular side effects. These interventions show how genetic diagnosis can guide customized treatment plans, particularly when standard pharmacotherapies and lifestyle changes have not worked.

## 4. Therapeutic Strategies, Challenges, and Future Directions

Syndromic obesity’s genetic complexity stems from numerous genes, transcription factors, and regulatory pathways controlling energy balance, hunger regulation, and metabolic rate. Many instances are still unexplained despite significant advancements, including the discovery of mutations in LEPR, MC4R, FTO, POMC, and SNORD116 and the creation of targeted medicines like Setmelanotide. Actionable mutations have been found in approximately 25–30% of early-onset severe obesity patients, and high-throughput genomic methods (WGS, WES, and NGS) have increased diagnostic yields in suspected instances to ~65% [63].

Ongoing challenges include the following:Regarding novel gene discovery, since many patients do not have a molecular diagnosis, single-cell analysis, multi-omics (epigenomics, transcriptomics, and metabolomics), and long-read sequencing are necessary to find uncommon or structural variations.Regarding phenotypic variability, patients with the same mutation may exhibit different expression, necessitating a move away from linear genotype–phenotype models and toward network-based methods.Regarding data sharing and standardization, to pool datasets, unify criteria, and validate variants of dubious importance, international cooperation is required.Regarding equitable diagnostics, in areas with limited resources, access to genomic testing is restricted; subsidized sequencing and scalable tele-genetics are required.Regarding next-generation therapies, in addition to MC4R agonists, newer alternatives include CRISPR-based gene editing, medicines that target the microbiome, and epigenetic modulators. Clinical studies use genotype-based patient classification.Addressing these gaps requires integrating genetic research, molecular diagnostics, and precision medicine within a globally coordinated framework.Additionally, it will be highly relevant to factor all these details into patient stratification in clinical trials that have yielded poor results [64]. The syndromic forms of obesity, though rare compared to common polygenic obesity, offer a unique window into the biological regulation of energy balance, adiposity, and neuroendocrine pathways. This review has comprehensively covered the genetic, clinical, and diagnostic landscape of syndromic obesity, bringing together a wide range of data sources, molecular mechanisms, and clinical phenotypes to offer a holistic understanding of the topic.

### 4.1. Future Directions in Research and Clinical Practice

Despite notable progress, multiple challenges and gaps remain, pointing to several promising avenues for future research. These future directions in syndromic obesity research and clinical management are visually summarized in Figure 5.
Novel Gene Discovery and Functional Validation: A significant number of individuals with strong clinical suspicion for syndromic obesity still receive no genetic diagnosis, suggesting the existence of unidentified genes or regulatory elements. Future research should adopt long-read sequencing, multi-omic integration [including transcriptomics, epigenomics, and metabolomics], and single-cell analysis to uncover rare or complex genomic variants.Gene–Environment and Epigenetic Interactions: The phenotypic variability observed among patients with the same genetic mutation [e.g., in LEPR or MC4R] suggests modulation by epigenetic changes and environmental factors. Investigating early-life exposures, maternal health, gut microbiota, and nutritional epigenomics could clarify these interactions and help predict phenotype severity.Standardization of Clinical and Diagnostic Protocols: There is an urgent need to establish standardized diagnostic algorithms for the early detection of syndromic obesity, especially in resource-limited settings where molecular diagnostics are not easily accessible. Incorporating clinical scoring tools based on features like developmental delay, dysmorphology, and metabolic profile can guide referrals for genetic testing.Expanding Access to Genetic Testing: Making molecular diagnostics affordable and accessible will ensure early intervention, appropriate therapy, and effective genetic counseling. Collaborative efforts between academic centers, public health systems, and international consortia are essential.Development of Targeted Therapies: Future therapeutic strategies should aim at personalized interventions, such as the following:
Gene-editing tools [e.g., CRISPR-Cas9].Targeted pharmacotherapy [e.g., receptor agonists].Modulation of epigenetic regulators or gut microbiota.Patient stratification based on genotype, which will be critical for designing effective clinical trials and ensuring treatment efficacy.
International Collaboration and Data Sharing: Given the rarity and diversity of these syndromes, international research collaborations are essential for pooling patient data, validating novel variants, and enhancing our understanding of pathogenic mechanisms.

The diagram above shows delineating prospective avenues in syndromic obesity research, encompassing the formulation of targeted therapies [gene editing and pharmacotherapy], innovative gene identification and validation via long-read sequencing and multi-omics, global collaboration for data exchange and variant verification, and the investigation of gene–environment interactions, including early-life exposures and maternal health. Other priorities encompass standardizing clinical practices through diagnostic algorithms and scoring systems and enhancing access to affordable genetic testing through joint initiatives.

Clinical studies concentrating on GLP-1/GIP receptor pharmacodynamics in syndromic obese populations and pharmacogenomic classification of individuals to forecast treatment responsiveness are examples of future directions. Moreover, combining these treatments with epigenetic targeting and gut microbiome manipulation may have synergistic advantages for individualized obesity treatment.

### 4.2. Limitations in Current Understanding

Even though significant improvements in understanding monogenic known syndromic forms of obesity have been made recently, many gaps still exist. For instance, novel causative genes are described, and phenotype–genotype redefinition is underway based on clinical ascertainment. On the other hand, the clinical complexity and the genetic and clinical heterogeneity underlying rare forms of syndromic obesity are hampering the obtaining of solid evidence in this context. The traditional approach of analyzing the sequence of single genes one by one is being replaced by multi-gene-related panels, whole-exome sequencing, or whole-genome sequencing to improve diagnosis yield. New advances in basic research have also been made, although a deeper understanding of the underlying mechanism is needed. Key limitations and recent advances in genetic diagnosis of syndromic obesity are illustrated in Figure 6.

This picture emphasizes the essential aspects contributing to comprehending genetic variation in obesity. This paradigm is centered on sophisticated genomic technologies, including whole-genome sequencing [WGS] and whole-exome sequencing [WES], which thoroughly investigate genetic variations. Multi-gene panels enable the concurrent examination of various obesity-related genes. The discovery of new causal genes enhances understanding of the genetic foundations of obesity. Furthermore, reinterpreting phenotype–genotype correlations elucidate the intricate interactions between genetic alterations and clinical manifestations. The picture recognizes the intrinsic clinical complexity of obesity, highlighting its multiple characteristics and the necessity to incorporate various genetic and phenotypic data for a more comprehensive understanding of its heterogeneity.

## 5. Conclusions

This review summarizes key genetic drivers, clinical presentations, and molecular diagnostic approaches in monogenic syndromic obesity. Significant gaps remain, including unidentified causative genes, incomplete genotype–phenotype correlations, and limitations in current molecular diagnostics. Future progress will depend on integrated genomic research, refined molecular diagnostics, and the development of precision therapies to improve outcomes for affected individuals.

## Figures and Tables

**Figure 1 cimb-47-00718-f001:**
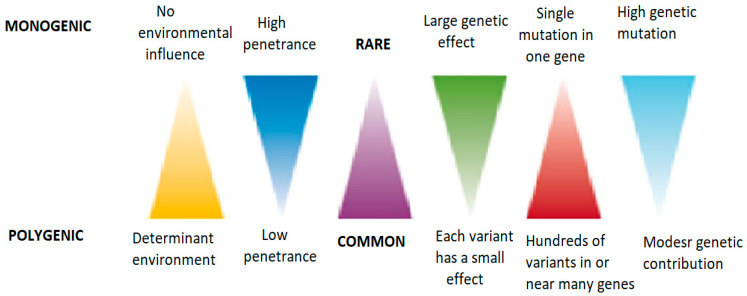
A comparison of monogenic and polygenic obesity regarding genetic influence, mutation type, and environmental contribution. Monogenic forms feature rare, high-impact mutations, whereas many common variants with minor effects cause polygenic forms.

**Figure 2 cimb-47-00718-f002:**
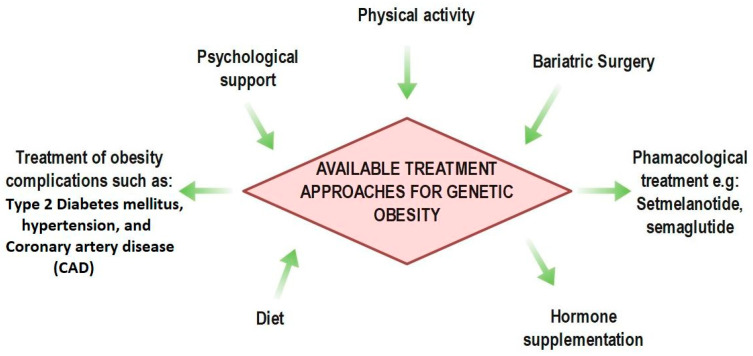
Multifaceted approaches to treating congenital or hereditary obesity include bariatric procedures, medication (e.g., Setmelanotide), lifestyle changes, and co-morbid disease therapy.

**Figure 3 cimb-47-00718-f003:**
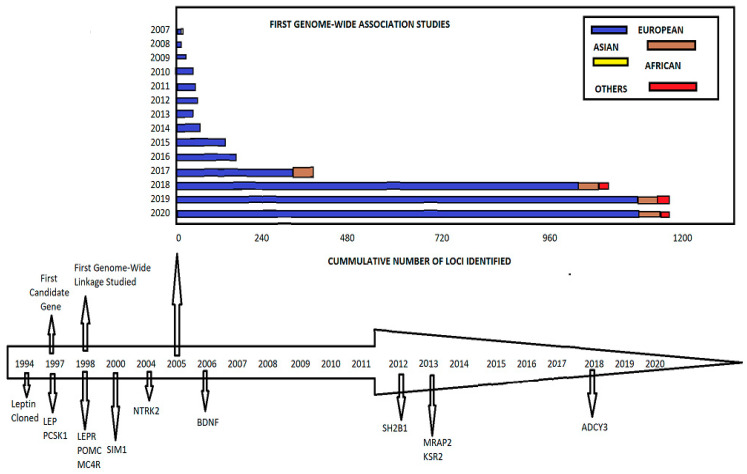
A timeline of important genetic discoveries in obesity, such as finding candidate genes and conducting genome-wide research. The bar graph illustrates the number of obesity-related loci found in different populations from 2007 to 2020.

**Figure 4 cimb-47-00718-f004:**
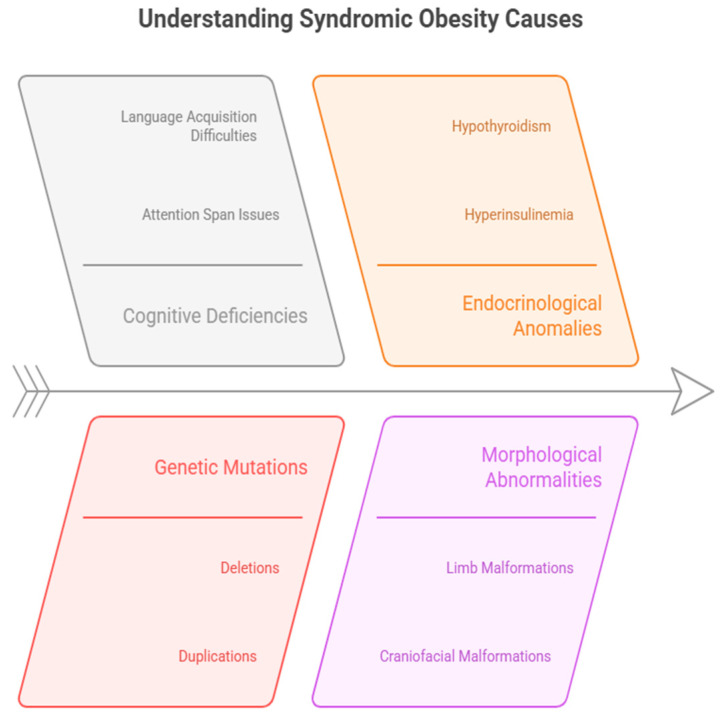
The intricate causes of syndromic obesity encompass cognitive, genetic, endocrinological, and morphological factors. Each category emphasizes particular abnormalities contributing to the disorder’s multifaceted nature.

**Figure 5 cimb-47-00718-f005:**
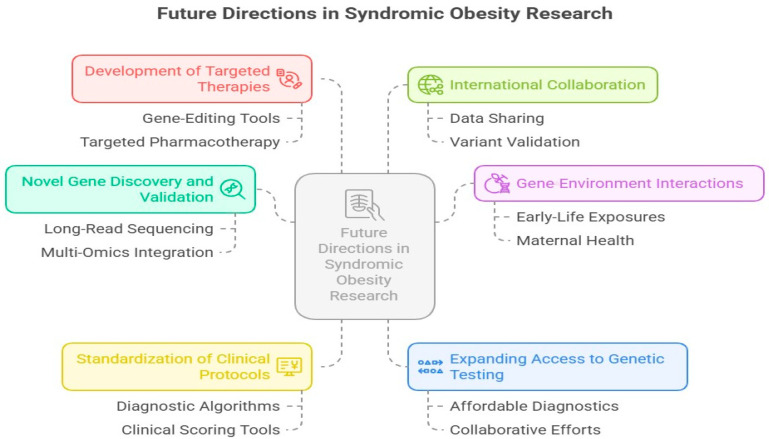
Future directions in syndromic obesity research focus on targeted therapies, gene discovery, international collaboration, and gene–environment interactions. Efforts also include standardizing clinical protocols and expanding access to affordable genetic testing.

**Figure 6 cimb-47-00718-f006:**
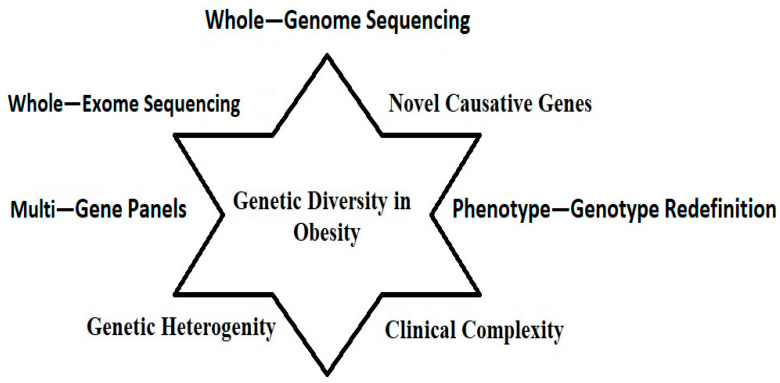
Limitations and advancements in genetic diagnosis of obesity: essential elements influencing our knowledge of genetic diversity in obesity, such as phenotype–genotype redefinition, novel gene discovery, and sophisticated sequencing methods.

**Table 1 cimb-47-00718-t001:** Major syndromic forms of obesity genetic etiology and core clinical features.

Syndrome	Genetic Etiology	Mode of Inheritance	Core Clinical Features
Prader–Willi Syndrome (PWS)	Loss of paternal 15q11–q13 (SNORD116 microdeletion or imprinting defect)	Paternal imprinting disorder	Neonatal hypotonia; hyperphagia; early-onset obesity; hypogonadism; intellectual disability; short stature; behavioral issues
Bardet–Biedl Syndrome (BBS)	Mutations in >20 BBS genes (e.g., BBS1, BBS10)	Autosomal recessive	Obesity; polydactyly; retinitis pigmentosa; renal anomalies; cognitive impairment; hypogonadism
Alström Syndrome	ALMS1 mutations	Autosomal recessive	Childhood obesity; cone–rod dystrophy; sensorineural hearing loss; insulin resistance; cardiomyopathy
WAGR Syndrome	Deletion at 11p13, including WT1 and PAX6	Sporadic (de novo)	Wilms tumor; aniridia; genitourinary anomalies; obesity; developmental delay
Cohen Syndrome	VPS13B gene mutations	Autosomal recessive	Truncal obesity; hypotonia; intellectual disability; retinal dystrophy; neutropenia
Carpenter Syndrome	RAB23 mutations	Autosomal recessive	Obesity; craniosynostosis; polydactyly; intellectual disability
Fragile X Syndrome	FMR1 CGG trinucleotide expansion	X-linked dominant	Macroorchidism; intellectual disability; autism features; hypotonia; obesity (in some males)
Borjeson–Forssman–Lehmann Syndrome (BFLS)	PHF6 mutations	X-linked recessive	Intellectual disability; hypogonadism; central obesity; gynecomastia; long philtrum
Albright Hereditary Osteodystrophy (AHO)	GNAS mutations affecting imprinting pattern	Maternal transmission	Short stature; round face; subcutaneous ossifications; obesity; hormone resistance (PTH)
Simpson–Golabi–Behmel Syndrome	GPC3 mutations	X-linked recessive	Pre- and postnatal overgrowth; organomegaly; obesity; developmental delay; coarse facial features
Beckwith–Wiedemann Syndrome (BWS)	Epigenetic abnormalities or paternal uniparental disomy at 11p15.5	Imprinting disorder	Macrosomia; macroglossia; neonatal hypoglycemia; embryonal tumors; obesity later in childhood
LEP/LEPR Deficiency	LEP or LEPR gene mutations	Autosomal recessive	Severe early-onset obesity; hyperphagia; hypogonadotropic hypogonadism; immune dysfunction
MC4R Deficiency	MC4R gene mutations	Autosomal dominant/recessive	Early-onset obesity; hyperphagia; tall stature; increased lean mass
POMC Deficiency	POMC gene mutations	Autosomal recessive	Hypocortisolism; red hair; hypoglycemia; early-onset obesity; hyperphagia
Alazami Syndrome	LARP7 gene mutations	Autosomal recessive	Microcephaly; short stature; facial dysmorphism; intellectual disability; obesity in adolescence
Craniopharyngioma-associated Hypothalamic obesity	Secondary to hypothalamic damage	Acquired	Rapid-onset obesity; impaired satiety; hypothalamic dysfunction; endocrine abnormalities

Abbreviations: BBS—Bardet–Biedl syndrome; AHO—Albright Hereditary Osteodystrophy; BFLS—Borjeson–Forssman–Lehmann syndrome; BWS—Beckwith–Wiedemann syndrome; LEP—Leptin gene; LEPR—Leptin receptor gene; MC4R—Melanocortin 4 receptor; POMC—Proopiomelanocortin; PTH—Parathyroid hormone.

## Data Availability

Not applicable.

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
