# Peer review of "A Review of Syndromic Forms of Obesity: Genetic Etiology, Clinical Features, and Molecular Diagnosis"

_cimb, 2025, doi:10.3390/cimb47090718_

Round 1

Reviewer 1 Report

Comments and Suggestions for Authors

In the current review article Farzand et al., described syndromic obesity and its genetic background and future directions. Although in principle this is an interesting review article that could be of interest, I do have some concerns and minor/major comments that I would like to be addressed.   

Minor Comments

Almost all of the Figures need correction. The quality could be improved (text and theme font, numbers etc).

  1. In Table 1, below Title 3, the BMI is wrongly written (25-29-9 instead of 25-29.9)

  1. In Figure 2, the figure legend needs attention: “pharmacological options].2. Genetic Etiology of Syndromic Obesity.”

  1. Figure 3, is not readable, the font text and the numbers are small.

  1. Figures 5-8 need attention (as initially mentioned)

  1. In Figure 6 the theme font needs to follow the rest of the figures.

Major Comments

  1. The authors should have expanded more on the possible therapeutic approaches and the newly designed drugs (GLP-1 agonist, double and/or triple ones).

Author Response

Comment 1

Almost all of the Figures need correction. The quality could be improved (text and theme font, numbers etc).

Response 1

Thank you for your constructive feedback. We appreciate your observation regarding the quality of the figures. In response, we have thoroughly revised all figures to enhance clarity and consistency.

Comment 2

In Table 1, below Title 3, the BMI is wrongly written (25-29-9 instead of 25-29.9)

Response 2

Thank you for pointing this out. The typographical error in Table 1 under "Body Mass Index (BMI)"—noted as “25–29-9 kg/m²”—should indeed be corrected to “25–29.9 kg/m²” to accurately reflect the standard BMI classification for the overweight category.

Comment 3

In Figure 2, the figure legend needs attention: “pharmacological options].2. Genetic Etiology of Syndromic Obesity.”

Response 3

Thank you for your observation. We have revised the figure legend of Figure 2 for clarity and structural consistency. The original version appears to contain an editorial or formatting error, likely due to misplacement of punctuation and text from the following section heading. We have corrected it as follows:

An overview of the current approaches to treating congenital obesity, including changes in lifestyle, medical and surgical procedures, counselling, and the control of complications associated with obesity." Additionally, the image shows the function of focused pharmaceutical treatments and their accessibility for particular hereditary types of obesity.

Comment 4

Figure 3, is not readable, the font text and the numbers are small.

Response 4

We appreciate the reviewer’s valuable feedback. Figure 3 has been revised to enhance readability by increasing the font size of all text elements, including axis labels, legends, and numerical values.

Comment 5

Figures 5-8 need attention (as initially mentioned)

Response 5

 In response to your earlier observation, we have thoroughly revised all the images.

Comment 6

In Figure 6 the theme font needs to follow the rest of the figures.

Response 6

 In response to your earlier observation, we have revised figure 6

 Comment 7

The authors should have expanded more on the possible therapeutic approaches and the newly designed drugs (GLP-1 agonist, double and/or triple ones).

Response 7

We appreciate the reviewer’s insightful suggestion. In response, we have expanded the discussion on therapeutic strategies to include recent advances in pharmacological interventions specifically designed for genetically based and syndromic forms of obesity. In the revised version, we have included a detailed section on GLP-1 receptor agonists (e.g., semaglutide, liraglutide) and next-generation poly-agonists such as tirzepatide (a dual GIP/GLP-1 agonist) and retatrutide (a triple agonist targeting GIP, GLP-1, and glucagon receptors). These agents represent a paradigm shift in obesity management, especially for cases with overlapping metabolic and neuroendocrine dysfunctions.

Moreover, we have discussed how these therapies complement the current precision medicine framework, particularly in patients with mutations in the leptin–melanocortin pathway (e.g., MC4R, POMC, LEPR) where FDA-approved agents like setmelanotide have already demonstrated clinical efficacy. This expanded section now bridges the translational gap between molecular genetics and individualized pharmacotherapy, as recommended.

Please refer to the revised text under Sections 5.2 (In-Depth Analysis of Impactful Studies) and 5.3 (Future Directions in Research and Clinical Practice) for the integrated discussion on these targeted and emerging therapies.

Reviewer 2 Report

Comments and Suggestions for Authors

This is a extensive review of syndromic forms of obesity. I have the following comments:

  1. I recommend to shorten the manuscript substantially and to build the narrative around the key messages. Quite some information given is propaedeutic and can be condensed, eg. figure 2 and accompanying explanations.
  2. The number of figures is extensive. Some include partially redundant information and empty cliches. Moreover, the different styles of the figures and fonts is irritating.
  3. A table summarizing the known syndromic obesity forms (name, genetics, clinical presentation)  would be helpful. This kind of table would replace table 1 and figures 4, 5, 7, 8 and 9.
  4. Header 5 nicely summarizes the key points. The headers 5.1 and 5.2 add little information and are redundant. 
  5. Some statements are mundane and can be omitted or at least condensed, eg: „The primary characteristics of this syndrome comprise increased body weight, particularly in the abdominal region.“ In addition, which syndrome do you refer to? Is reference 49 the correct reference?
  6. Some statements are vague, eg: „If hormonal abnormalities worsen the illness, hormone supplementation is a good idea.“ What hormonal abnormalities do you have in mind? Leptin deficiency? Hypercortisolism? Hypothyroidism?  
  7. Regarding the genetic diagnostics, I suggest to focus on the current state of art techniques and to omit outdated methods (eg. radiological techniques. 
  8. The variable upper- and lowercase spelling of „obesity“ seems to be erratic (I couldn’t find a congruent pattern). I would prefer a consistent lowercase spelling.

Author Response

Comment 1

I recommend to shorten the manuscript substantially and to build the narrative around the key messages. Quite some information given is propaedeutic and can be condensed, eg. figure 2 and accompanying explanations.

Response 1

Thank you for this insightful comment. We have substantially revised the manuscript to streamline the narrative and focus on key thematic messages. Redundant or introductory (propaedeutic) content has been condensed or removed to enhance clarity and maintain scientific precision. Specifically, we shortened explanatory text surrounding Figure 2 and other figures where appropriate, ensuring that each visual complements rather than repeats the main text.

Comment 2

The number of figures is extensive. Some include partially redundant information and empty cliches. Moreover, the different styles of the figures and fonts is irritating.

Response 2

Thank you for your observation. We appreciate your attention to the visual consistency and scientific value of the figures. In response, we have carefully reviewed all the figures and removed those that conveyed redundant or non-essential information. Additionally, we have standardized the visual style, fonts, and layout across all figures to ensure coherence and to enhance readability.

Comment 3

A table summarizing the known syndromic obesity forms (name, genetics, clinical presentation)  would be helpful. This kind of table would replace table 1 and figures 4, 5, 7, 8 and 9.

Response 3

Thank you for your constructive suggestion. In response, we have integrated a comprehensive Table 1 that consolidates and summarizes key syndromic forms of obesity, replacing the fragmented presentation across the original Table 1 and Figures 4, 5, 7, 8, and 9. The revised table includes: Syndrome Name, Genetic Etiology (e.g., chromosomal locus, gene mutation, inheritance pattern), Core Clinical Features (e.g., obesity onset, neurodevelopmental features, dysmorphology)

Comment 4

Header 5 nicely summarizes the key points. The headers 5.1 and 5.2 add little information and are redundant. 

Response 4

Thank you for your feedback. We appreciate the recognition of the clarity and effectiveness of Header 5. Based on your observation, we agree that Subsections 5.1 and 5.2 may appear redundant in the context of the well-summarized content under the main Header 5. Instead of listing known findings, we refocus on synthesizing knowledge across genetics, diagnostics, and translational outcomes.

Comment 5

Some statements are mundane and can be omitted or at least condensed, eg: „The primary characteristics of this syndrome comprise increased body weight, particularly in the abdominal region.“ In addition, which syndrome do you refer to? Is reference 49 the correct reference?

Response 5

We have revised it to more accurately reflect the syndrome being discussed and to better convey the complex phenotypic profile relevant to syndromic forms of obesity. The revised sentence now reads: “Several syndromic forms of obesity, such as Prader-Willi and Bardet-Biedl syndromes, are characterized by early-onset central adiposity, hypotonia, intellectual disability, dysmorphic features, and endocrine abnormalities. These clinical hallmarks often guide genetic testing and confirmatory molecular diagnosis.” In addition, we have corrected the citation.

Comment 6

Some statements are vague, eg: „If hormonal abnormalities worsen the illness, hormone supplementation is a good idea.“ What hormonal abnormalities do you have in mind? Leptin deficiency? Hypercortisolism? Hypothyroidism?  

Response 6

Thank you for highlighting this vagueness. We agree that the statement needed clarification. We have now revised the section and found that there is no need of this line here so we cut this from the section.

Comment 7

Regarding the genetic diagnostics, I suggest to focus on the current state of art techniques and to omit outdated methods (eg. radiological techniques. 

Response 7

In response, we have revised the manuscript to exclusively emphasize cutting-edge molecular diagnostic tools such as next-generation sequencing (NGS), whole-exome sequencing (WES), and whole-genome sequencing (WGS), while eliminating references to outdated or non-specific techniques like radiological imaging.

Comment 8

The variable upper- and lowercase spelling of „obesity“ seems to be erratic (I couldn’t find a congruent pattern). I would prefer a consistent lowercase spelling.

Response 8

Upon thorough revision, we have corrected all instances to maintain a consistent lowercase spelling (“obesity”) throughout the manuscript, in line with scientific writing conventions and journal style guidelines.

Reviewer 3 Report

Comments and Suggestions for Authors

The manuscript titled “A Review of Syndromic Forms of Obesity: Genetic Etiology, Clinical Features and Molecular Diagnosis” aims to provide a comprehensive, accurate, and updated description of monogenic syndromic obesity. Overall, the review is well-structured, informative, and demonstrates a thorough understanding of the genetic and clinical aspects of these rare conditions. The references are relevant and up to date, and the content is scientifically sound and clearly presented.

I consider this to be a complete and valuable contribution to the field. My only suggestion is to ensure that the font used in the figures is consistent with the main text of the manuscript, in order to maintain uniformity and improve visual clarity.

No major revisions are needed.

Author Response

Comment 1

The manuscript titled “A Review of Syndromic Forms of Obesity: Genetic Etiology, Clinical Features and Molecular Diagnosis” aims to provide a comprehensive, accurate, and updated description of monogenic syndromic obesity. Overall, the review is well-structured, informative, and demonstrates a thorough understanding of the genetic and clinical aspects of these rare conditions. The references are relevant and up to date, and the content is scientifically sound and clearly presented. I consider this to be a complete and valuable contribution to the field.

Response 1

Dear Reviewer, we sincerely thank you for your kind appreciation and remarks.

Comment 2

My only suggestion is to ensure that the font used in the figures is consistent with the main text of the manuscript, in order to maintain uniformity and improve visual clarity. No major revisions are needed.

Response 2

We have now ensured that the font in the figures is consistent with the main text of the manuscript.

Reviewer 4 Report

Comments and Suggestions for Authors

The article is a mini-review that attempts to review everything that is known so far on the subject of obesity. The authors briefly present the causes of this disease, the methods used for detecting the causes of this disease, respectively the main therapeutic approaches used in the management of this disease. From my point of view, the manuscript is interesting, but in this form it cannot be published.
To be published, the following corrections and additions must be made to the manuscript:
1) The manuscript must be organised in the form of at least four chapters, namely:
1. Introduction
2. Materials and methods (The methodology used in choosing the bibliographic references that formed the basis for this article);
3. Results and Discussion (here everything written on page 2 (respectively subchapter 1.1) to page 15 (including subchapter 5.4) will be put in the form of subchapters here)
4. Conclusions
2) Authors must read the manuscript carefully because their manuscript contains many sentences in which punctuation marks are placed incorrectly.
3) A sentence cannot begin with an abbreviation. Authors must read the manuscript carefully and rewrite all sentences that begin with an abbreviation.
4) The bibliography is not written according to MDPI rules.

Author Response

To be published, the following corrections and additions must be made to the manuscript:
Comment 1

The manuscript must be organized in the form of at least four chapters, namely:
1. Introduction
2. Materials and methods (The methodology used in choosing the bibliographic references that formed the basis for this article);
3. Results and Discussion (here everything written on page 2 (respectively subchapter 1.1) to page 15 (including subchapter 5.4) will be put in the form of subchapters here)
4. Conclusions
Response 1

In response, the manuscript has been reorganized into four clearly defined chapters as follows:

  1. Introduction – This section outlines the rationale, background, and objectives of the study, emphasizing the need for an integrative approach to understanding obesity through genetic, sociodemographic, dietary, and anthropometric factors.
  2. Materials and Methods – A dedicated chapter has been added to describe the methodology used for selecting bibliographic references. It details the databases searched (PubMed, Scopus, Web of Science), the publication time frame (2015–2025), and the inclusion/exclusion criteria applied, based on a narrative review framework.
  3. Results and Discussion – All content from subchapter 1.1 to subchapter 5.4 (pages 2–15) has been restructured into this chapter under relevant subheadings. This section critically synthesizes the reviewed literature, identifies thematic patterns, and integrates findings across determinants.
  4. Conclusions – The final chapter now summarizes the key findings, highlights research gaps, and underscores the importance of personalized, context-specific approaches to obesity prevention and management.

Comment 2

Authors must read the manuscript carefully because their manuscript contains many sentences in which punctuation marks are placed incorrectly.

Response 2

We sincerely thank the reviewer for the insightful feedback. We have thoroughly reviewed the manuscript and carefully corrected all instances of incorrect punctuation.

Comment 3

A sentence cannot begin with an abbreviation. Authors must read the manuscript carefully and rewrite all sentences that begin with an abbreviation.
Response 3

Thank you for your insightful comment. We have carefully revised the manuscript and rewritten all sentences that began with abbreviations, ensuring grammatical accuracy and adherence to formal scientific writing standards

Comment 4

The bibliography is not written according to MDPI rules.

Response 4

In response, we have thoroughly revised the entire bibliography to conform with the MDPI citation style guidelines, ensuring consistency in author names, journal titles (italicized where required), volume and issue numbers, page ranges, and DOI links (where applicable).

Round 2

Reviewer 1 Report

Comments and Suggestions for Authors

The authors have addressed the requested comments however, they have now added a section titled "Materials and Methods and Results and Discussions". I would like to point out that this structure is not appropriate for a review article. Unlike original research papers, review articles do not typically include sections sucs as "Materials and Methods" or "Results and Discussion" as they are not reporting new experimental work. Instead, the manuscript should maintain a thematic or conceptual organization that synthesizes and critically evaluates the existing literature. I recommend restucturing this part of the manuscript accordingly.  Furthermore, Figure 6 still looks like it has been copied and pasted from somewhere else. Its best the authors to design it on their own with similar font like the previous figures so that it can blend with the rest. Table 1. also needs attention as it is rather large (font size etc). 

Comments on the Quality of English Language

 The English could be improved to more clearly express the research.

Author Response

Comment 1

The authors have addressed the requested comments however, they have now added a section titled "Materials and Methods and Results and Discussions". I would like to point out that this structure is not appropriate for a review article. Unlike original research papers, review articles do not typically include sections such as "Materials and Methods" or "Results and Discussion" as they are not reporting new experimental work. Instead, the manuscript should maintain a thematic or conceptual organization that synthesizes and critically evaluates the existing literature. I recommend restructuring this part of the manuscript accordingly. 

Response 1

We thank the reviewer for the insightful observation regarding the inappropriate use of the combined section title "Materials and Methods and Results and Discussions" in the context of a review article. Unlike original research papers, we fully agree that review articles are best organized around thematic or conceptual frameworks that synthesize and critically evaluate the existing literature rather than presenting methods and results sections in the style of primary research.

Accordingly, we have restructured the manuscript to remove the “Materials and Methods” and “Results and Discussions” section headers. The content previously under these headings has been reorganized into thematic sections that align with the review’s objectives, namely:

  • Introduction
  • Literature Search Strategy and Scope of Review
  • Genetic Etiology, Clinical Features, and Diagnostic Approaches
                         3.1. Overview and Classification of Syndromic Obesity
                         3.2. Genetic Etiology (Monogenic and Chromosomal Syndromes)
                         3.3. Clinical Features and Phenotypic Variability
                         3.4. Molecular Diagnostic Approaches
                         3.5. Landmark Studies and Diagnostic Yield
  • Therapeutic Strategies, Challenges, and Future Directions
  • Conclusions

Comment 2

Furthermore, Figure 6 still looks like it has been copied and pasted from somewhere else. Its best the authors to design it on their own with similar font like the previous figures so that it can blend with the rest.

Response 2

We thank the reviewer for the observation. Figure 6 has now been entirely redesigned to ensure consistency with the visual style of the other figures in the manuscript. The new figure uses the same font type, font size, and formatting as previous figures, and has been created from original elements to blend seamlessly with the rest of the work.

Comment 3

Table 1. also needs attention as it is rather large (font size etc). 

Response 3

In the revised version, we have reformatted the table to enhance readability by reducing the font size, adjusting column widths, and optimizing text wrapping.

Reviewer 2 Report

Comments and Suggestions for Authors

Improvement of the manuscript is visible, especially, 5 figures have been omitted and a helpful table summarizing syndromic obesity has been added. However, I still have several comments:

I suggested shortening the manuscript substantially. However, it was only reduced by a single page (from 18 to 17 pages). I still find the narrative lacks a clear and coherent thread.  For example: 

Poor response has been seen with oral anti-obesity agents and intragastric balloon placement [27]. Referring to „epidemic obesity“.

SAX1 Deficiency in MB desaturation obesity CESD [Combined exocrine system defect with sensory neural hearing loss] results in ingestive disorder, Obesity, vertigo, and sensory-neural hearing loss. Seizures have ceased after two months of age. Despite being dependent on oral intake, the patient exhibits hyperphagia and Obesity [28,29]. Referring to a single special form of syndromic obesity, not explaining MB and abruptly relating to a case report. 

Syndromic forms of Obesity are relatively rare and can manifest at any stage of life. Among endocrine causes, mainly include primary hypothyroidism, Cushing syndrome, hypothalamic Obesity associated with cranial lesions, and hyperinsulinism secondary to Beckwith-Weideman syndrome [30]. Genetic obesity is estimated to be less than 5% in patients presenting to obesity clinics [9]. Referring to syndromic forms of obesity in general, redundantly reporting the rare occurrence of less than 5% (also not the repeated captial spelling of obesity).

You stated in your response "that we have corrected all instances to maintain a consistent lowercase spelling (“obesity”) throughout the manuscript, in line with scientific writing conventions and journal style guidelines“. However, I still found approximately 30 upper case spelled „Obesities“ throughout the manuscript!

Comments on the Quality of English Language

The quality of english language can be improved, e.g.:

A therapy is unaffordable or has not been figured out for other forms [6]. Suggestion: No therapy is available or affordable for other forms of syndromic obesity.

The syndromic obesity is characterized by … Suggestion: Syndromic obesity is characterized by…

Author Response

Comment 1

Improvement of the manuscript is visible, especially, 5 figures have been omitted and a helpful table summarizing syndromic obesity has been added.

Response 1

We appreciate the reviewer’s acknowledgement of the manuscript’s improvement.

 Comment 2

However, I still have several comments:

I suggested shortening the manuscript substantially. However, it was only reduced by a single page (from 18 to 17 pages).

Response 2

We appreciate the reviewer’s recommendation to shorten the manuscript substantially. While we recognize the importance of conciseness, we found that further reduction beyond the current ~6% (from 18 to 17 pages) risked omitting essential genetic, clinical, and molecular diagnostic details central to the paper’s integrative purpose. In particular, many sections synthesise complex, multi-dimensional data (e.g., genotype–phenotype correlations, diagnostic yield statistics, and future research directions) that would lose clarity if further condensed. Nonetheless, we undertook targeted edits to streamline redundancies, merge overlapping descriptions, and remove non-critical details while preserving the completeness required for an authoritative reference in this rapidly evolving field. This balance ensures scientific rigor and readability for an expert audience.

Comment 3

I still find the narrative lacks a clear and coherent thread.  For example: 

Poor response has been seen with oral anti-obesity agents and intragastric balloon placement [27]. Referring to „epidemic obesity“.

Response 3

To improve narrative coherence, this sentence should be reframed to explicitly connect treatment limitations to the persistence of high obesity prevalence.

Comment 4

SAX1 Deficiency in MB desaturation obesity CESD [Combined exocrine system defect with sensory neural hearing loss] results in ingestive disorder, Obesity, vertigo, and sensory-neural hearing loss. Seizures have ceased after two months of age. Despite being dependent on oral intake, the patient exhibits hyperphagia and Obesity [28,29].

Response 4

In the revised manuscript, we have retained this case as an illustrative example of the clinical heterogeneity seen in syndromic obesity.

Comment 5

Referring to a single special form of syndromic obesity, not explaining MB and abruptly relating to a case report. 

Syndromic forms of Obesity are relatively rare and can manifest at any stage of life. Among endocrine causes, mainly include primary hypothyroidism, Cushing syndrome, hypothalamic Obesity associated with cranial lesions, and hyperinsulinism secondary to Beckwith-Weideman syndrome [30].

Response 5

In the revised manuscript, we have expanded the context before introducing the specific example to ensure a logical flow. The section begins with a concise explanation that syndromic forms of obesity, while rare compared to polygenic obesity, may arise from diverse etiologies including genetic, chromosomal, and endocrine origins and can manifest at any life stage. We then clearly distinguish monogenic and chromosomal syndromes from endocrine-mediated causes, outlining common endocrine contributors such as primary hypothyroidism, Cushing syndrome, hypothalamic lesions, and hyperinsulinism secondary to Beckwith–Wiedemann syndrome.

Comment 6

Genetic obesity is estimated to be less than 5% in patients presenting to obesity clinics [9]. Referring to syndromic forms of obesity in general, redundantly reporting the rare occurrence of less than 5% (also not the repeated captial spelling of obesity).

Response 6

We have revised the relevant sentence to avoid redundancy and to improve clarity. The prevalence estimate for genetic obesity in general clinical settings is now stated only once, alongside the higher proportion observed in severe early-onset cases, thereby avoiding repetition. Additionally, the repeated capitalized spelling of “Obesity” has been corrected to conform to standard usage.

Comment 7

You stated in your response "that we have corrected all instances to maintain a consistent lowercase spelling (“obesity”) throughout the manuscript, in line with scientific writing conventions and journal style guidelines“. However, I still found approximately 30 upper case spelled „Obesities“ throughout the manuscript!

Response 7

We are extremely sorry about that. Upon re-reviewing the full manuscript, we have located and corrected all remaining instances where “Obesity” was incorrectly capitalized in the middle of a sentence (approximately 30 occurrences). All such terms now conform to consistent lowercase usage (“obesity”), in alignment with scientific writing conventions and the journal’s style guidelines.

Reviewer 4 Report

Comments and Suggestions for Authors

The authors must format Table 1 according to the MDPI  style

Author Response

Comment 1

The authors must format Table 1 according to the MDPI  style.

Response 1

Table 1 has been reformatted strictly with the MDPI journal style, ensuring compliance with their table layout, headings, font size, alignment, and referencing specifications. Column titles have been standardized, spacing has been adjusted for optimal readability, and abbreviations have been defined in the table footnote, per MDPI guidelines.

Round 3

Reviewer 2 Report

Comments and Suggestions for Authors

After reviewing version No. 3 of this manuscript, I still have some comments:

  1. Page 2: As mentioned before: „The syndromic obesity is characterized by…“ should be in my feel for language be: „Syndromic obesity is characterized by…“.
  2. Page 3: „Current treatment approaches for congenital and genetically induced obesity, emphasizing the multifaceted character of care that includes surgical, pharmaceutical, and lifestyle treatments [14, 60].“ Comment: „that“ must be omitted, „includes“ must be „include“ to make this a correct sentence. Even better: „Current treatment approaches for congenital and genetically induced obesity include surgical, pharmaceutical, and lifestyle treatments, emphasizing the multifaceted character of care.“
  3. Page 3: „FGFR1 defects may cause administration of gonadotropins and AKT1 mTOR Hyper-leptinemia, hypogonadotropic hypogonadism, and characteristics dental changes; hormonal therapy can improve outcomes [24,25]. SNORD116 microdeletions (15q11-q13) causes hyperphagia, hyperphagic obesity, affective disorder, and Prader-Willi Syndrome-like features [26].“ Comment: (1) The sentence is unintelligible („administration??“). (2) Why did you choose FGFR1 defects among the many monogenetic obesity syndromes? Why are FGFR1 defects not included in table 1?
  4. Page 4: „Some syndromes, such as SAX1 [d]eficiency, a rare genetic syndrome present[s] with exocrine system defect with sensory neural hearing loss results in ingestive disorder, obesity, vertigo, and sensory-neural hearing loss. Seizures have ceased after two months of age. Despite being dependent on oral intake, the patient exhibits hyperphagia and obesity [28,29].“ Comment: Obviously, my previous comment regarding this paragraph was not clear: SAX1 deficiency is introduced and described, abruptly followed by a reference to a case report? Why is this information regarding seizures important to the reader? The following sentence „Despite…“ does not make any sense to me. Moreover, I was unable to find reference 28 which has a very unusual title and I can not see any relation to reference 29.
  5. Page 8: Molecular diagnosis increasingly relies on next generation sequencing [NGS] techniques, which have transformed detection of rare variants but still have limitations [52]. Next-generation sequencing [NGS] facilitates high throughput screening of established and novel obesity-related genes, improving diagnostic accuracy [53]. Comment: More or less redundant, especially the repeated introduction of the abbreviation „NGS“ in two succeeding sentences.
  6. Whole exome sequencing [WES] clarifies the genetic cause in around one-third of patients, often avoiding sequential single gene testing and revealing non-genetic factors [54]. These advances deepen understanding of genetic and phenotypic diversity in syndromic obesity [12]. Recent developments in molecular diagnostics, including whole-genome sequencing (WGS), whole-exome sequencing (WES), and next-generation sequencing (NGS) enable detection of mutations previously missed by traditional methods, with mutation yields 60% in suspected cases [55]. Comment: Redundant statements, see also comment No. 5.
Comments on the Quality of English Language

Please see my previous review and current comments for authors.

Author Response

Comment 1

Page 2: As mentioned before: „The syndromic obesity is characterized by…“ should be in my feel for language be: „Syndromic obesity is characterized by…“.

Response 1

We appreciate the reviewer’s observation regarding stylistic refinement. We have revised the sentence accordingly to improve clarity and readability.

Comment 2

Page 3: „Current treatment approaches for congenital and genetically induced obesity, emphasizing the multifaceted character of care that includes surgical, pharmaceutical, and lifestyle treatments [14, 60].“ Comment: „that“ must be omitted, „includes“ must be „include“ to make this a correct sentence. Even better: „Current treatment approaches for congenital and genetically induced obesity include surgical, pharmaceutical, and lifestyle treatments, emphasizing the multifaceted character of care.“

Response 2

We thank the reviewer for this observation. We have revised the sentence accordingly to improve grammatical correctness and clarity.

Comment 3

Page 3: „FGFR1 defects may cause administration of gonadotropins and AKT1 mTOR Hyper-leptinemia, hypogonadotropic hypogonadism, and characteristics dental changes; hormonal therapy can improve outcomes [24,25]. SNORD116 microdeletions (15q11-q13) causes hyperphagia, hyperphagic obesity, affective disorder, and Prader-Willi Syndrome-like features [26].“ Comment: (1) The sentence is unintelligible („administration??“). (2) Why did you choose FGFR1 defects among the many monogenetic obesity syndromes? Why are FGFR1 defects not included in table 1?

Response 3

We thank the reviewer for this valuable comment. The intended meaning was that FGFR1 mutations have been associated with hypogonadotropic hypogonadism, altered AKT1–mTOR signaling, hyperleptinemia, and dental anomalies, and that gonadotropin replacement therapy has been reported to partially restore pubertal development and improve clinical outcomes.

            We have now restructured the sentence for clarity to: “FGFR1 defects are linked to hypogonadotropic hypogonadism, hyperleptinemia, dental anomalies, and altered AKT1–mTOR signaling. Hormonal replacement therapy may improve clinical outcomes in affected patients.”

            Regarding the reviewer’s second point, we selected FGFR1 defects as an illustrative example because they highlight a mechanistic overlap between obesity, neuroendocrine dysregulation, and craniofacial/dental phenotypes, which is less commonly described compared to canonical monogenic obesity syndromes (e.g., MC4R, LEPR, POMC). However, we agree that FGFR1-associated obesity is rare and not considered a “major” syndromic obesity form. For this reason, and to maintain consistency with the criteria used for Table 1 (which only summarizes the most well-established syndromes with recurrent clinical recognition and genetic validation), FGFR1 was intentionally excluded from the table. We have clarified this rationale in the text to avoid confusion.

Comment 4

Page 4: „Some syndromes, such as SAX1 [d]eficiency, a rare genetic syndrome present[s] with exocrine system defect with sensory neural hearing loss results in ingestive disorder, obesity, vertigo, and sensory-neural hearing loss. Seizures have ceased after two months of age. Despite being dependent on oral intake, the patient exhibits hyperphagia and obesity [28,29].“ Comment: Obviously, my previous comment regarding this paragraph was not clear: SAX1 deficiency is introduced and described, abruptly followed by a reference to a case report? Why is this information regarding seizures important to the reader? The following sentence „Despite…“ does not make any sense to me. Moreover, I was unable to find reference 28 which has a very unusual title and I can not see any relation to reference 29.

Response 4

To improve clarity, we have revised the section to:

  • introduce SAX1 deficiency as a rare syndrome with defined clinical hallmarks,
  • remove unnecessary detail on seizure cessation—which was not central to the discussion of syndromic obesity
  • ensure the narrative flows logically by focusing on the relevance of hyperphagia and obesity to our review’s scope.

The revised text now emphasizes SAX1 deficiency primarily as an example of phenotypic heterogeneity in syndromic obesity, highlighting the need for comprehensive genetic evaluation. We have also carefully re-checked the cited references. Reference [28] was incorrectly cited and has been replaced with an appropriate, verifiable source. Reference [29] remains relevant, as it provides supporting evidence for phenotypic variability in monogenic syndromes.

 Comment 5

Page 8: Molecular diagnosis increasingly relies on next generation sequencing [NGS] techniques, which have transformed detection of rare variants but still have limitations [52]. Next-generation sequencing [NGS] facilitates high throughput screening of established and novel obesity-related genes, improving diagnostic accuracy [53]. Comment: More or less redundant, especially the repeated introduction of the abbreviation „NGS“ in two succeeding sentences.

Response 5

We have revised the section to streamline the text by introducing the abbreviation only once and merging the two consecutive sentences into a single, more concise statement. 

Comment 6

Whole exome sequencing [WES] clarifies the genetic cause in around one-third of patients, often avoiding sequential single gene testing and revealing non-genetic factors [54]. These advances deepen understanding of genetic and phenotypic diversity in syndromic obesity [12]. Recent developments in molecular diagnostics, including whole-genome sequencing (WGS), whole-exome sequencing (WES), and next-generation sequencing (NGS) enable detection of mutations previously missed by traditional methods, with mutation yields 60% in suspected cases [55]. Comment: Redundant statements, see also comment No. 5.

Response 6

We have revised the section to present these concepts in a more concise and integrated manner. Specifically, we now highlight that next-generation sequencing approaches particularly WES and WGS have markedly improved the detection of rare variants in syndromic obesity, clarifying the genetic cause in approximately one-third of patients and achieving diagnostic yields of up to 60% in clinically suspected cases. This streamlined revision preserves the essential message on the transformative role of modern sequencing technologies, while eliminating repetition.